# Genomic characterisation of an entomopathogenic strain of *Serratia ureilytica* in the critically endangered phasmid *Dryococelus australis*

Joanne L. Allen[1]*, Nicholas P. Doidge[1,2], Christina Cheng[1,2], Michael Lynch[2], Helen K. Crabb[1], Jean-Pierre Scheerlinck[3], Rhys Bushell[4], Glenn F. Browning[1], Marc S. Marenda[1,4]

1 Asia-Pacific Centre for Animal Health, Melbourne Veterinary School, Faculty of Veterinary and Agricultural Sciences, The University of Melbourne, Parkville, Victoria, Australia, 2 Melbourne Zoo, Parkville, Victoria, Australia, 3 Centre for Animal Biotechnology, Melbourne Veterinary School, Faculty of Veterinary and Agricultural Sciences, The University of Melbourne, Parkville, Victoria, Australia, 4 Melbourne Veterinary School, Faculty of Veterinary and Agricultural Sciences, The University of Melbourne, Werribee, Victoria, Australia

☯ These authors contributed equally to this work.
* jlallen@unimelb.edu.au

**Data Availability Statement:** The data is available at the following repositories. Serratia ureilytica AM923 Bioproject Accession: PRJN666536; Assembly: GCA_022559505.1; BioSample:

## Abstract

Between 2014 and 2019, unexpected mortalities were observed in a colony of *Dryococelus australis*, an endangered stick-insect kept at the Melbourne Zoo for a breeding and conservation program. Pure cultures of *Serratia* spp. were obtained from the haemolymph of moribund and recently deceased individuals. The combined bacteriological and histopathological observations suggested an infectious cause of these mortalities. Genotyping of *Serratia* sp. isolated from the insects and their environment revealed a predominant strain profile. A representative isolate, AM923, was entirely sequenced and compared to 616 publicly available *Serratia* spp. genomes, including 37 associated with insects. The genomes were distributed into 3 distinct groups, with 63% of the insect-associated isolates within a single clade (clade A) containing AM923, separated from most environmental/plant-associated strains (clade B) and human isolates (clade C). Average nucleotide identity and phylogenetic analyses identified AM923 as *S. ureilytica* and revealed similarities with putatively entomopathogenic strains. An experimental infection model in honey bees (*Apis mellifera*) confirmed the pathogenic potential of AM923. A urease operon was found in most insect isolates and a PCR assay, based on the *ure*B gene sequence, was used to confirm the presence of AM923 in experimentally infected bees. This species-specific PCR could be applied to detect entomopathogenic *Serratia* spp. in infected insects or their environment.

SAMN16287482 GenBank Accessions: CP070508.1 Serratia ureilytica strain AM923 chromosome, complete sequence CP070509.1 Serratia ureilytica strain AM923 plasmid unnamed1, complete sequence CP070510.1 Serratia ureilytica strain AM923 plasmid unnamed2, complete sequence Serratia marcescens AM1004 Bioproject Accession: PRJN666537 Assembly: GCA_020252365.1 BioSample: SAMN16287483" https://www.ncbi.nlm.nih.gov/nuccore/NZ_JAFFPY000000000.1.

**Funding:** The author(s) received no specific funding for this work.

**Competing interests:** The authors have declared that no competing interests exist.

## Introduction

The Lord Howe Island stick insect (*Dryococelus australis*) is a critically endangered phasmid (Order Phasmatodea, also known as Phasmida) which was endemic to Lord Howe Island, a small volcanic island located between Australia and New Zealand in the Southern Pacific Ocean. The insect was assumed to be extinct following the incursion of the European black rat (*Rattus rattus*) onto the island in the early 20[th] century. A small number of *D. australis* were discovered on a nearby island in 2001 and some individuals were transported to Melbourne Zoo to establish a breeding colony. This colony now contains almost 500 insects [1].

In recent years, increased mortalities have occurred in this captive population. Major mortality episodes commencing in 2013 prompted histopathological and microbiological investigations into the causes of the deaths. Histopathology investigations of dead and moribund insects between October 2013 and March 2015 revealed inflammatory lesions throughout the body in 97% of all cases, with aggregates of short Gram negative rods frequently found within lesions. Bacteria from the genus *Serratia* were isolated from the haemocoel in more than 80% of sampled individuals and often in pure culture [2]. Haemolymph is a normally sterile site and pure growth of bacteria on a rich non-selective culture medium, such as sheep blood agar, in high numbers is deemed to be clinically important.

The genus *Serratia* contains 32 species, including *S. marcescens*, *S. ureilytica* and *S. liquefaciens* [3]. Many *Serratia* spp. are saprophytic organisms with the ability to survive in various environmental conditions and to utilise a broad range of nutrients; they have been isolated from water, soil and plants, as well as vertebrate and invertebrate animals [4]. *Serratia* spp. are also nosocomial opportunistic pathogens that have been implicated in a wide variety of infections in humans, including urinary tract, respiratory tract and wound infections, meningitis, endocarditis and septicaemia [5], mostly in immunocompromised patients and in neonates [6, 7]. The presence of multiple antimicrobial resistances in many isolates is particularly concerning [8]. The molecular epidemiology of *Serratia* spp. outbreaks has been investigated by pulsed-field gel electrophoresis (PFGE) after genomic cleavage with SpeI, and this approach has been found to yield good levels of reproducibility and discrimination [6, 9–12]. No multi-locus sequence typing (MLST) scheme has been developed to date for this genus and sequencing of the 16s rRNA gene does not provide sufficient discriminatory power to characterise isolates. Other molecular typing methods have been used to characterise isolates, most recently a high throughput short sequence typing scheme (HiSST) to trace human opportunistic infections in a clinical setting [13].

*Serratia* spp. can infect insects belonging to various orders, including Orthoptera, Isoptera, Coleoptera, Lepidoptera, Hymenoptera, Diptera and Phasmatodea [4, 7, 14]. Experimental inoculation of *Serratia* spp. into the haemocoel can lead to septicaemia and death in more than 70 insect species [15], including *Drosophila melanogaster* [16]. *Serratia* spp. have been isolated from diseased wild mole crickets (*Scapteriscus borellii*) [17] and from larvae and adults of the European honey bee (*Apis mellifera*) [7]. Variations in pathogenicity have been documented in experimentally infected honey bees, with strain KZ-19 found to be a highly virulent isolate compared to strain sicaria Ss1 [18, 19]. Some *Serratia* spp. and nematodes can have mutually beneficial relationships that result in entomopathogenic properties [20]. Plants found in the insect digestive tract are a potential source of *Serratia* spp., but the route of entry from the gut into the haemocoel is not well understood. Rupture of the gut has been suggested as a possible route of entry [7]. In addition to horizontal transmission, vertical transmission of *Serratia* spp. via insect eggs has been demonstrated in other insect species, including *Heliothis virescens* and *Heliothis zea* [21–23]. Colonisation of both the external egg surface and the egg contents has been demonstrated and found to be associated with a reduction in egg production and hatching rates [22].

The objectives of this study were: (1) Characterisation of *Serratia* spp. associated with recently deceased and moribund insects of the captive *D. australis* colony of Melbourne Zoo; (2) Determination of the complete genomic sequences of 2 representative strains and comparison with publicly available genomes of *Serratia* spp., including entomopathogenic strains; (3) Assessment of the virulence potential of the predominant isolate in experimentally infected honey bees (*Apis mellifera*).

## Materials and methods

### Sample collection from *D. australis*

Necropsies were conducted on *D. australis* within the Melbourne Zoo colony that died or were found moribund and euthanised between August and October 2014, and between November 2016 and May 2017. Haemolymph samples were collected from the haemocoel and inoculated onto sheep blood agar (SBA) and MacConkey agar (MAC). The inoculated plates were incubated at 37°C overnight. Histopathological examinations were conducted on a subset of insects to compare the lesions in insects from which: 1) *Serratia* spp. were isolated in pure culture; 2) *Pseudomonas* sp. or *Proteus* sp. were cultured; and 3) there was no significant bacterial growth. The lesions were assessed by Christine Bayley, Gribbles Pathology, using a previously published scoring system [2]. The variances of the lesion scores for the 3 groups were compared using the Kruskal-Wallis test.

### Sample collection from the insect environment

Environmental samples were collected from free ranging captive insect enclosures in February 2015. Four types of samples were collected: nest boxes, floor surfaces, drinking water, and frass (insect excrement). Outer surfaces of each nest box and sections of the floor (1 m$^2$) that were free of frass and plant material were swabbed with sterile pre-moistened gauze swabs (7.5 cm$^2$). Plant material refers to potted plants or fresh clippings of tree lucerne or tagasaste (*Chamaecytisus prolifer*), Lord Howe Island tea tree (*Melaleuka howeana*), Morton Bay fig (*Ficus macrophylla*), holly oak (*Quercus ilex*), *Baloghia* sp. and *Pittosporum* sp. provided as a food source. Swabs were placed in sterile saline (10 mL) and agitated for 30 seconds. Drinking water (10 mL) was collected from each water dish. Pooled frass samples were collected from inside each nest box and from the floor directly below. Frass pellets were homogenised, suspended in sterile saline (0.1 g/10 mL) and left to settle for 30 minutes. All processed samples were cultured on LB agar containing cycloheximide (0.125 mg/mL), an anti-fungal agent, and incubated overnight at 37°C.

### Bacterial isolation and identification

*Serratia* spp. colonies were identified based on the following phenotypic characteristics: oxidase negative, Gram negative rods, negative for lactose and arabinose fermentation and production of indole and positive for acetoin production, citrate utilisation and evidence of DNase activity. Gram negative bacilli that were oxidase positive were presumptively identified as *Pseudomonas* or *Aeromonas* spp. If swarming growth was evident, the colony was identified as *Proteus* spp. Other colonies containing organisms that were Gram positive or that were morphologically distinct from *Serratia* spp. were classified as "other bacteria".

### Molecular typing

For the initial molecular typing analyses, *Serratia* spp. from haemolymph samples that yielded a pure growth were inoculated into tryptone soya broth (Oxoid) and incubated overnight at

37˚C. Genomic DNA was prepared as described previously [24] and digested with the restriction endonuclease, SpeI. PFGE was performed using the CHEF-DRâ III System (Bio-Rad Laboratories, Hercules, California, USA) under the following conditions: 1% agarose gel (0.5% TBE), 14˚C for 20 hours, at 6.0 V/cm, with a 120˚ included angle, and initial and final switch times of 2.2 and 54.2 seconds, respectively. Gels were stained with ethidium bromide (0.5 mg/mL) for 30 minutes, destained in distilled water and photographed using a Molecular Imager® ChemiDoc XRS⁺ imaging system. Fingerprinting rep-PCRs [25, 26] were performed using the ThermoPolâ DNA polymerase (0.2 units/20 μL reaction volume) and 1x buffer (New England Biolabs), 1 mM dNTPs, 0.4 μM of each of the primers ERIC1 (5′-ATGTAAGCT CCTGGGGATTCAC-3′) and ERIC2 (5′-AAGTAAGTGACTGGGGTGAGCG-3′), and approximately 50 ng of genomic DNA template. The reaction conditions: an initial denaturation at 95˚C for 2 minutes, 30 cycles of denaturation at 95˚C for 20 seconds, annealing at 50˚C for 30 seconds and extension at 68˚C for 3.5 minutes, with a final extension step of 5 minutes at 68˚C. Amplicons were analysed by electrophoresis in 2% agarose gels in TAE buffer followed by staining with GelRed® (Biotium). Dendrograms were constructed from PFGE and rep-PCR data using GelJ [27]. The PFGE band patterns were analysed for similarity using the Dice method, and the unweighted pair group method with arithmetic mean (UPGMA), with a 3.0% tolerance level, for linkage. The rep-PCR band patterns were analysed using Pearson curve-based similarity coefficients and the UPGMA linkage method, with a 1% tolerance level, as suggested previously [10].

## Whole genome sequencing

For Illumina sequencing, *S. ureilytica* AM923 and *S. marcescens* AM1004 genomic DNAs were extracted with the High Pure PCR Template Preparation Kit (Roche), purified and concentrated in a final elution volume (30 μL) with the DNA Clean and Concentrator kit (Zymo), and stored at -20˚C. The quality of the genomic DNA preparations was assessed by gel electrophoresis in a 1% agarose gel in 0.5 x TBE buffer (45 mM Tris, 45 mM borate, 1.0 mM EDTA, pH 8.3) containing SYBR Safe DNA gel stain (Invitrogen). Hyperladderä 1kb (Bioline) DNA molecular weight markers were used to determine DNA fragment size and visually assess DNA integrity. The extracted DNAs were quantified using a Qubit® 3.0 Fluorometer (Life Technologies) with the dsDNA BR kit, and the RNA/DNA ratios were checked using a Nanodropä (v3.8.1) spectrophotometer (Thermo Fisher Scientific). Genomic DNA quality testing, library preparation using the Nextera XT DNA library kit v3.0 and Illumina short read sequencing of the extracted DNA were conducted in accordance with the manufacturer's instructions by the Australian Genome Research Facility. Reads were filtered for quality scores >20 and adapters were removed using the wrapper script Trim Galore v0.4.4 [28].

For Nanopore sequencing, *S. ureilytica* AM923 genomic DNA was purified using the DNeasyâ kit on a QiaCube Extractor (Qiagen) and cleaned by solid phase reversible immobilisation with Agencourt AMPureX magnetic beads. The quality of the genomic DNA preparation was assessed with a Nanodrop spectrophotometer and quantified with a Quantusä fluorometer using the QuantiFluorâ dsDNA reagent (Promega). Genomic DNA was sheared with a Covaris gTube to generate ~8 kb fragments. The sequencing library was prepared with the Oxford Nanopore kit SQK-LSK208, using a mixture of 2.2 μg of sheared DNA and 0.55 μg of unsheared DNA. Sequencing was performed with a MinION MK-Ib device fitted with a Flowcell R9.4 for 17 hours. Base calling was performed using Albacore version 1.1.2 to generate 2D reads in FASTQ format. Reads were filtered for length of >1 kb and mean PHRED quality of >15 with the script fastq_to_fastq.py [29].

## Sequence analysis

Genome assemblies were performed with Unicycler v0.4.0 [30] using both Illumina and Nanopore reads for strain AM923 (hybrid assembly), and Illumina reads only for strain AM1004. Genome sequence data for *S. ureilytica* AM923 and *S. marcescens* AM1004 have been deposited at the NCBI under the BioProject numbers PRJNA666536 and PRJNA666537, respectively. Assembly FASTA files were annotated with Prokka [31]. Prophages were identified with the online tool, PHASTER [32]. The number and location of integrative conjugative elements (ICE) were predicted with ICEberg online server [33]. Genomic islands were explored with the online tool IslandViewer 4 [34]. The assembled genomes were screened for antimicrobial resistance and virulence genes with ABRicate [35].

Whole genome phylogenetic analysis of strains AM923 and AM1004 was performed with the program REALPHY [36] against complete sequences of *Serratia* spp. (S1 Table), using *S. marcescens* strain ATCC_13880 as reference. The REALPHY sequence alignment output was converted into a phylogenetic tree with RaxMLv8.2.11 [37] using the GTR model of nucleotide substitution with the Gamma parameter. Average Nucleotide Identity (ANI) analysis of strains AM923 and AM1004 was performed against representative genomes of *Serratia* spp. with the program FastANI [38]. Pan-genome analysis was performed with the Roary pan genome pipeline [39] with default settings (minimum identity for BlastP of 95%, core genes possessed by at least 99% isolates), on partial or complete *Serratia* spp. genomes (S2 Table) downloaded from the NCBI assembly database and systematically reannotated with Prokka [31] for consistency. For phylogenetic analyses, the entire set of CDSs from the *Serratia* spp. genomes mentioned above were searched for a selection of core or housekeeping genes, using the corresponding AM923 CDSs as Blastn query sequences. Individual genes with at least 70% identity with and 90% coverage of each query CDS were extracted and compiled into multi-FASTA files. Genomes that did not contain a full set of query sequences were discarded from the analysis. Multiple alignments were constructed separately for each multi-FASTA file with the program MUSCLE [40], then concatenated into a single alignment file. Identical sequences were removed from the alignment to construct unrooted phylogenetic trees with RaxML v8.2.11 [37] using the GTR model of nucleotide substitution and the Gamma parameter. Trees were visualised and annotated with the Forester phylogeny decorator program [41] or the Interactive Tree of Life online tool [42].

## Virulence assay in honey bees

*Serratia ureilytica* AM923 and *Escherichia coli* K12 were grown to mid-log phase in LB at 37˚C, harvested by centrifugation and resuspended in sterile phosphate buffered saline, pH 7.4 (PBS) (137 mM NaCl, 2.7 mM KCl, 10 mM $Na_2HPO_4$ and 1.8 mM $KH_2PO_4$) to final concentrations of 1.12 10e8 cfu/mL and 3.7 10e7 cfu/mL, respectively. The inoculum was a 1:1 mixture of the bacterial suspension and sucrose/water (equal parts sucrose and water). Honey bees (*Apis mellifera*) were collected from a bee hive located on the University of Melbourne campus and were experimentally infected by the immersion method, as described previously [18]. Briefly, bees were separated into groups of approximately 30 insects, placed in sterile tubes (50 mL) and the inoculum (500 μL) was added to each tube. A 1:1 mixture of sterile PBS and the sucrose/water mix was used for the control. To enable screening for the presence of *Serratia* spp., randomly selected bees were removed and euthanised by chilling to -80˚C prior to the experiment.

For each treatment group, insects from 2 tubes were mixed into a cup cage (500 mL), in duplicate (S1 Fig). Bees were maintained at 30˚C and fed 50% sucrose solution and sterile water over the course of the trial. The numbers of dead bees were recorded daily, and cadavers were removed and placed at -80˚C. After a period of 8 days, the surviving bees were counted

and euthanised by chilling to -80˚C. The survival curves were analysed with the R packages survival (https://cran.r-project.org/web/packages/survival) and survminer.

The abdomens of dead bees were placed in individual sterile microtubes (1.5 mL) and crushed with a disposable tissue grinder pestle (Axygen) in sterile PBS (100 μL). A sample of this homogenate (50 μL) was plated directly on MacConkey agar using the streak-dilution technique for a semi-quantitative culture. Identification of *Serratia* spp. was based on the following phenotypic characteristics: oxidase negative, Gram negative rods, positive for gelatin liquefaction, DNase activity and the presence of urease after incubation at 37˚C for 48 hours. Total DNA was extracted with the Wizardâ Genomic Purification kit (Promega) using the Animal Tissue protocol and diluted in nuclease-free water for PCR assays. This procedure was used to test the insects removed prior to the experiment for the carriage of *Serratia* spp..

### PCR assays

A 67 bp fragment of the *Serratia* spp. nuclease gene *nuc*A was amplified with the primers nucA_F (5′-CAATGTGTCGATCGTGCGTC-3′) and nucA_R (5′-CCAGTTGGCGAAT TTGGTGG-3′). A 77 bp fragment of the *Serratia* spp. urease subunit beta gene *ure*B was amplified with the primers ureB_F (5′-CGAGATTGAGGTGGCGCTTA-3′) and ureB_R (5′-CCCAACCGTCCACCAGATTA-3′). All reactions were performed using the Thermopolâ Taq polymerase (0.2 units/20 μL reaction) in 1 x buffer (New England Biolabs), with 1 mM of each dNTP and 0.4 μM of each primer, using an initial denaturation incubation at 95˚C for 3 minutes, then incubation through 30 cycles of denaturation for 20 seconds at 95˚C, annealing for 30 seconds at 55˚C, and elongation for 40 seconds at 68˚C, with a final extension incubation for 5 minutes at 72˚C. Amplicons were visualised after electrophoresis in a 3% agarose gel stained with Gel Redâ (Biotium).

## Results

### Predominance of *Serratia* spp. infections during elevated mortalities in a captive population of *D. australis*

A total of 140 recently deceased or moribund *D. australis* were examined and 127 haemolymph samples were collected for bacteriological investigation. Non-pigmented strains of *Serratia* spp. were detected in 73 (57.5%) samples; of those, 47 (71.2%) were obtained in pure culture. In contrast, *Proteus* sp. and *Pseudomonas* sp. were isolated in pure culture from 5 and 6 cases, respectively (Table 1). No growth was observed for 10 (7.8%) samples. Seven isolates (2 non-pigmented and 5 red pigmented) of *Serratia* spp. were also detected in environmental samples collected from the drinking water, floors, nest boxes and insect frass (S3 Table).

Histopathological analysis of a subset of 25 animals found those from which *Serratia* spp. were isolated in pure culture had significantly higher lesion scores for the head and fat body (P < 0.05) than those from which no significant growth was obtained. No difference was seen between insects from which *Pseudomonas* sp./*Proteus* sp. were isolated and the other groups of insects (S4 Table).

**Table 1. Bacteriological isolation rates from insect haemolymph samples.**

| | Pure cultures (n = 66) | Mixed cultures (n = 51) | Mixed cultures with *Serratia* spp. | Mixed cultures without *Serratia* spp. |
|---|---|---|---|---|
| *Serratia* spp. | 47 (71.2%) | 26 (50.9%) | - | - |
| *Proteus* sp. | 5 (7.6%) | 11 (21.6%) | 6 (11.8%) | 5 (9.8%) |
| *Pseudomonas* sp. | 6 (9.1%) | 16 (31.4%) | 6 (11.8%) | 10 (19.6%) |
| Other bacteria | 8 (12.1%) | 42 (82.4%) | 20 (39.2%) | 22 (43.1%) |

## Typing of the isolates from *D. australis* suggests the persistence of a complex *Serratia* spp. population associated with the insect colony

PFGE analysis of 24 insect and 5 environmental isolates of *Serratia* spp. defined 2 pulso-types, designated 'Type A' and 'Type B', each further subdivided into subtypes 1 and 2 (Fig 1 and S5 Table). Profile analysis showed that Type A was predominant, representing almost 80% of the strains—21/24 insect isolates and 2/5 environmental isolates. Of the Type A isolates, 12 were subtype A1 and 8 were subtype A2, with a single outlier (AM1003). All Type A isolates were >95% similar and were non-pigmented (Fig 2). Subtype A1 contained only insect isolates, while subtype A2 contained two of the environmental isolates (5W2w, 6F1w). Type B strains had 74% similarity with Type A strains and showed a greater pheno-typic heterogeneity; subtype B1 comprised 3 non-pigmented insect isolates (VW348,

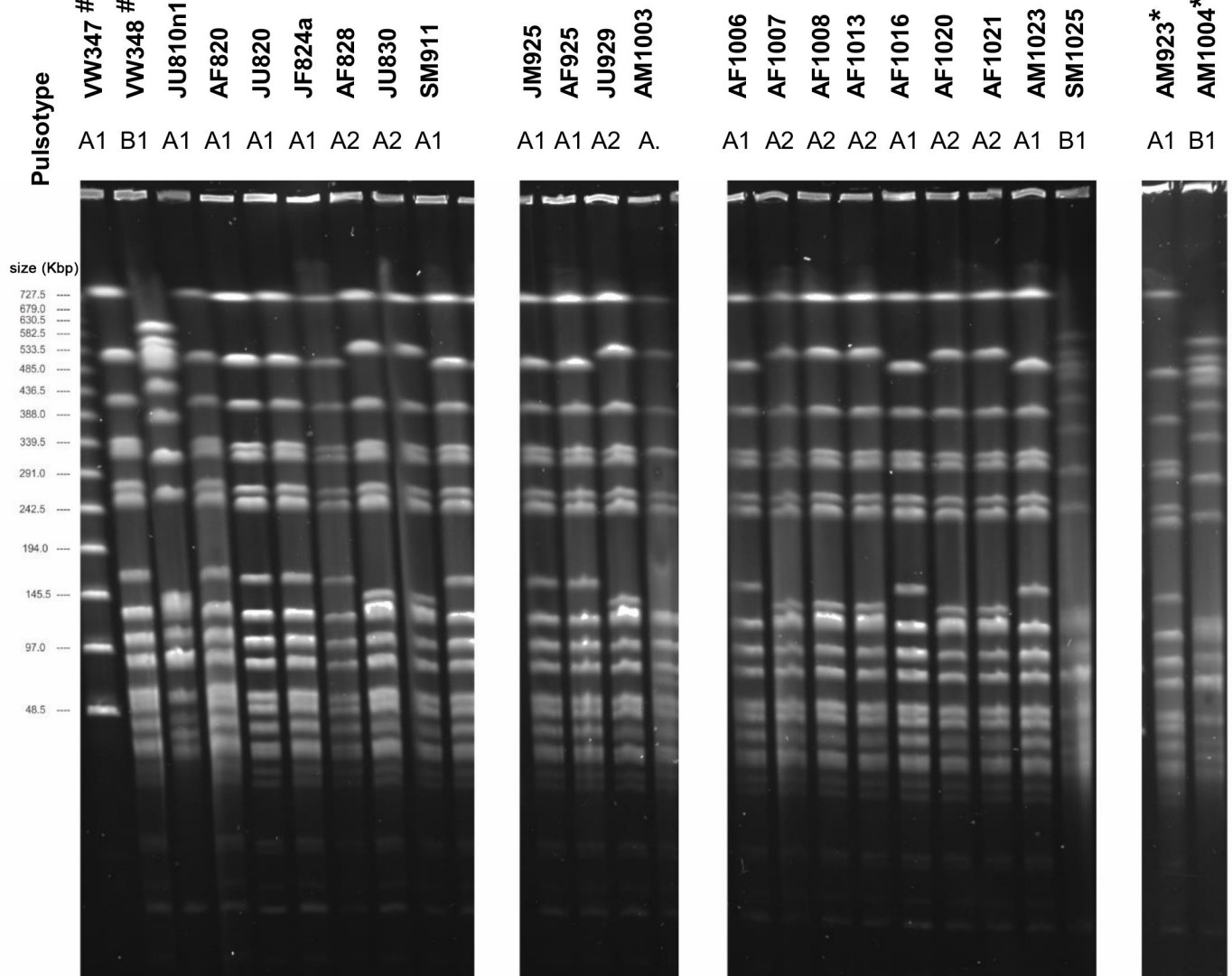

**Fig 1. Pulsed field gel electrophoresis analysis of *Serratia* spp. isolated from haemolymph of captive *D. australis* between August and October 2014.** # Isolates (VW347 and VW348) from insects collected prior to this study and provided by Zoos Victoria. *Strains representative of the pulsotypes A and B and selected for complete genome sequencing. Molecular weight marker: lambda PFG ladder (New England Biolabs).

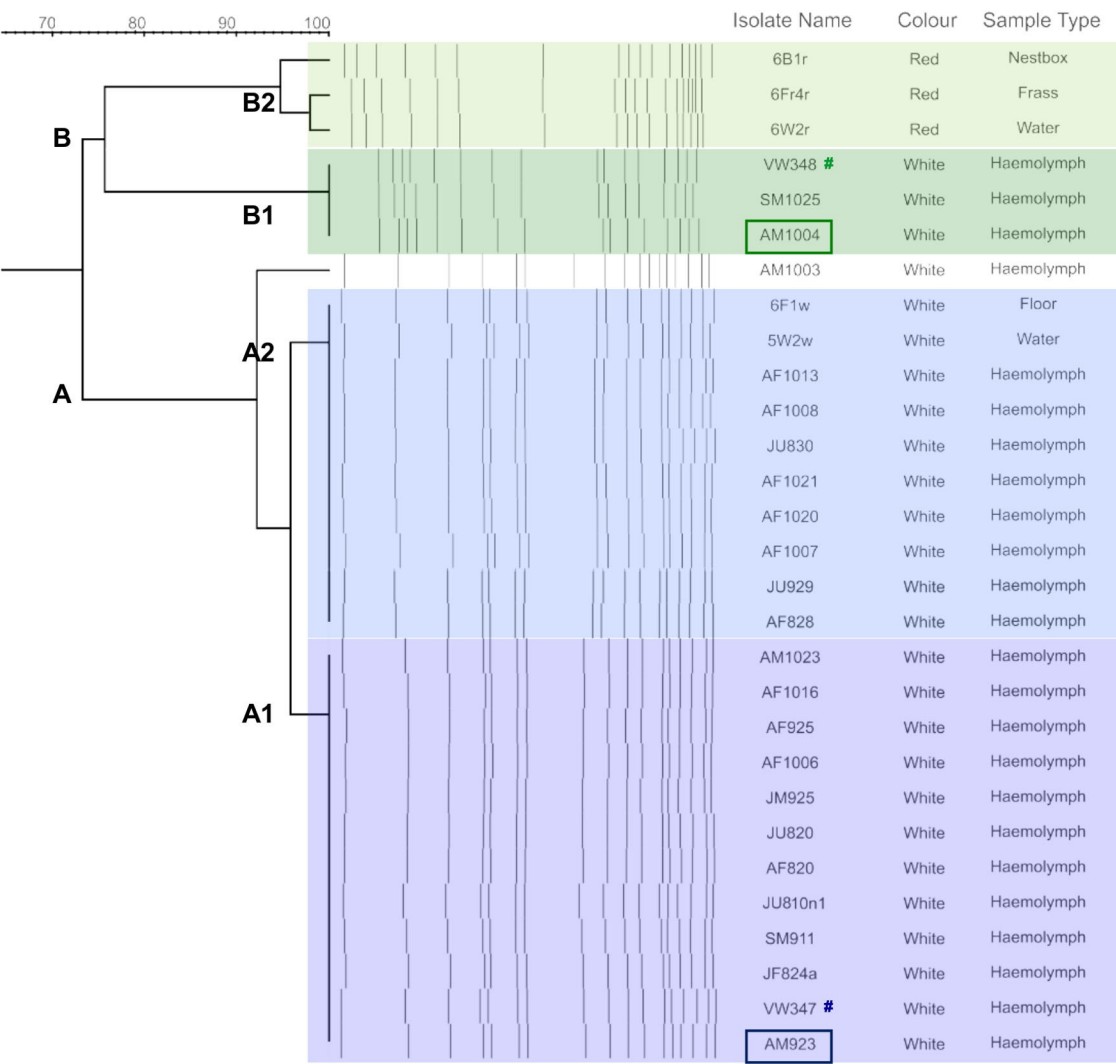

**Fig 2. Dendrogram of PFGE profiles, pigmentation phenotypes and isolation sources.** Twenty two *Serratia* spp. isolated from *D. australis.* between August and October 2014 and *Serratia* spp. isolated from the insect environment in February 2015 were compared. Vertical bars represent restriction endonuclease cleavage fragments. # VW347 and VW348 represent isolates from insects collected prior to this study and provided by Zoos Victoria. Boxes indicate the two representative isolates selected for genome sequencing.

SM1025 and AM1004), and subtype B2 comprised 3 pigmented environmental isolates (6W2r, 6B1r and 6Fr4r).

To evaluate the significance of the predominance of PFGE Type A strains within Melbourne Zoo, fingerprinting rep-PCR was performed on a panel of the more recent *Serratia* spp. isolates collected from insects, as well as unrelated strains from Australian companion animals (dogs, cats, horses, a rabbit and a bird) and from infected *D. australis*, originally sourced from the Melbourne Zoo colony and kept at two international zoos (Bristol and Toronto). Profile analysis of the amplicons broadly supported previous PFGE observations and indicated that most strains isolated from dead or moribund *D. australis* at the Melbourne Zoo between 2017 and 2019 were highly similar to the dominant Type A isolated between 2013 and 2014 (S2 Fig and S5 Table). A rep-PCR profile, very similar to the one yielded by Type A strains, was also detected for an isolate from the *D. australis* colony in Toronto Zoo.

Apart from the strains 2008–163 and 2016–324, isolated from a bird and a cat, respectively, isolates from domestic animals and from the Bristol Zoo appeared to be unrelated to the Melbourne Zoo strains (S2 Fig and S5 Table).

## Sequence analysis of *Serratia* sp. AM923 and comparison with other isolates

Two non-pigmented isolates cultured from dead insects, AM923 and AM1004, representing the dominant Type A and minor Type B1 strains, respectively, were selected for genome sequencing (Table 1). The strain AM923 sequence was chosen as the main reference genome for in-depth analysis, while strain AM1004 was used for comparative purposes. The genome of strain AM923 was completely assembled into a 5,215,760 bp chromosome and two cryptic plasmids of 38329 bp and 6891 bp. The chromosome was predicted to contain one Integrative Conjugative Element (ICE) and 3 prophages. Antimicrobial resistance genes were detected on the chromosome and were predicted to confer resistance to chloramphenicol, fluoroquinolones, cefotaxime, tetracyclines, aminoglycosides and macrolides. No antimicrobial resistance genes were found on the plasmids. The genome of strain AM1004 was partially assembled into 30 contigs (N50 = 550,042 bp, longest segment 2,004,609 bp), representing a total of 5,108,759 bp.

Whole-genome polymorphism analysis of 131 completely sequenced isolates from the genus *Serratia* with the program REALPHY placed strains AM923 and AM1004 into two distinct subsets of *S. marcescens*, while some other isolates fell into more distant groups on the phylogenetic tree (Fig 3). The clade containing strain AM923 also contained two *Serratia ureilytica* strains, as well as two undefined *Serratia* species. In contrast, strain AM1004 fell into a clade composed of only *S. marcescens*. Species identification by ANI analysis (S6 Table)

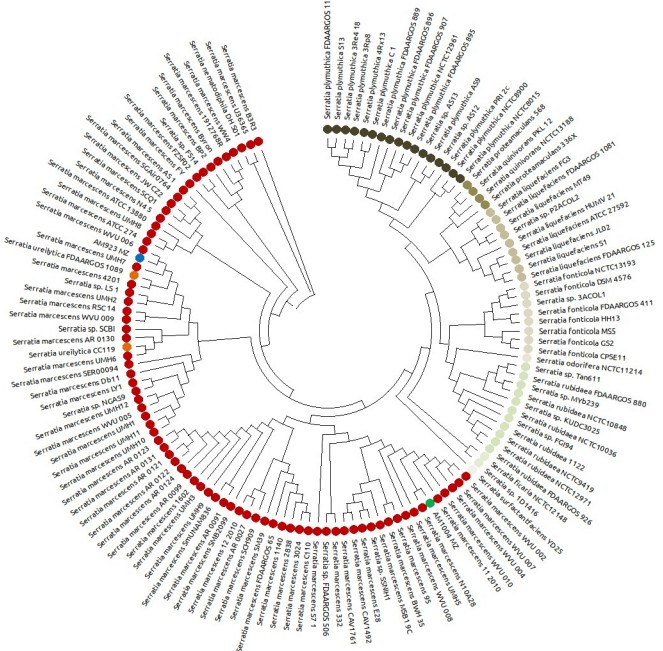

**Fig 3. Cladogram of 131 complete *Serratia* spp. genomes.** An unrooted tree was constructed from the alignment of polymorphisms produced by REALPHY using RAxML (GTR model and Gamma parameter). Red nodes: *S. marcescens*, orange nodes: *S. ureilytica*, other *Serratia* species: shades of brown-green. *D. australis* isolates AM923 and AM1004 are indicated in blue and green, respectively.

strongly supported the classification of AM923 as *S. ureilytica* (99.0%), while AM1004 was tentatively classified as *S. marcescens* (95.09%).

To explore the phylogenetic position of these two isolates and confirm the REALPHY analysis and ANI results, a pan-genome analysis of 708 complete or partial *Serratia* spp. genomes, including *S. ficaria*, *S. fonticola*, *S. grimesii*, *S. liquefaciens*, *S. marcescens*, *S. nematodiphila*, *S. odorifera*, *S. plymuthica*, *S. proteamaculans*, *S. quinivorans*, *S. rubidaea*, *S. symbiotica and S. ureilytica* (S2 Table), was performed to find core genes suitable for phylogenetic analysis at the whole genus level. This analysis identified eight sequences found in at least 99% of strains across all species within the genus, namely *aro*K, *rpl*M, *met*J, *rps*I, *pts*H, *lpr*1, *gln*K and *inf*A, which were compiled into a concatenated multiple sequence alignment representing the 650 genomes carrying this set of core genes. The alignment included 438 identical sequences (i.e., a redundant set of concatenated genes without a polymorphic position among different genomes), which were removed from the dataset, and a phylogenetic tree was built from the remaining 209 unique sequences. Within this tree, most species of *Serratia* were well separated from each other and all *S. marcescens* isolates fell into a large sub-tree divided into branches containing predominantly strains from either human or non-human sources (S3 Fig). The isolates AM923 and AM1004 were placed in two distant branches of the *S. marcescens* sub-tree, confirming the REALPHY results. The *S. marcescens* sub-tree also contained 5 *S. nematodiphila* and 3 *S. ureilytica*, placed in branches mainly composed of non-human isolates. The *S. ureilytica* isolates were all in the sub-tree containing AM923.

To refine this analysis and better ascertain the positions of strain AM923 and strain AM1004, individual trees were constructed from 14 non-paralogous housekeeping genes expected to be present in all *S. marcescens* genomes, namely *aro*K, *par*E, *aro*C, *adk*, *rec*R, *dna*J, *pur*A, *rpo*H, *rho*, *gyr*B, *rpo*B, *dna*A, *dna*K, and *uvr*A. A concatenated multiple alignment of 616 sequences was obtained, from which 294 identical sequences were removed. The resultant phylogenetic tree of the remaining 322 unique sequences contained three large clades (Fig 4), mainly composed of strains isolated from insects (clade A, bootstrap support value 100%), the environment or plants (clade B, bootstrap support value 52%), and humans, blood or hospitals (clade C, bootstrap support value 97%).

While insect isolates accounted for only 6% of the total 616 *Serratia* spp. genomes investigated, they represented almost 40% of clade A. In contrast, insect isolates were markedly less prevalent in clades B and C (Table 2). Strain AM923 fell into clade A, along with isolates from various other insects, including a moth (*Orthaga achatina*, strain LS-1), a mole cricket (*Scapteriscus borellii*, strain ADJS-2D-white), and a honey bee (*Apis mellifera*, strain KZ-19). Other *Serratia* spp. isolated from insects in the order Hemiptera (*Orius laevigatus* and *Orius niger*), as well as a low-virulence bee pathogen, strain sicaria Ss1, were also present in this group. Moreover, clade A contained at least 3 *Serratia ureilytica* and 1 *Serratia nematodiphila*, consistent with our earlier phylogenetic analysis at the genus level. Strain AM1004 was placed in clade C, which harboured most human-associated isolates, as well as 3 strains from mosquitoes (Fig 4).

## Development of PCR assays for detection of entomopathogenic *Serratia* spp. based on genome analysis

Virulence genes or sequence markers that could potentially detect entomopathogenic *Serratia* spp. in clinical specimens and/or environmental samples were investigated by comparative genome analysis. The gene presence/absence output file from the Roary analysis of 708 *Serratia* spp. genomes, which clustered proteins with at least 90% sequence similarity, was searched for sequences preferentially associated with insect isolates. Sixty-three protein coding sequences (CDSs) with a predicted function present in all insect isolates from clade A (which includes

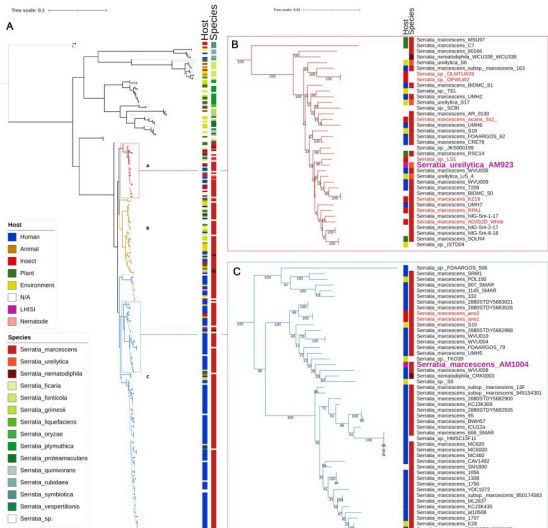

**Fig 4. Phylogenetic analysis of concatenated alignments of 14 housekeeping genes from 616 genomes of the genus *Serratia*.** Panel A: mid-rooted tree built from 322 non-identical sequences, including 256 *S. marcescens*, *S. nematodiphila* and *S. ureilytica* isolates, with RAxML and the GTR + Gamma substitution model. Bootstrap support values were inferred from 100 replicates and represented by branch thickness on the complete tree. Branches are drawn to scale and colour-coded according to the clade: red, clade A; brown-green, clade B; blue, clade C. Nodes are colour-coded according to the origin of the strain. Insets showing the groups containing the Melbourne Zoo isolates AM923 (Panel B) and AM1004 (Panel C) with bootstrap support values >50% indicated on the branches. The scale bars indicate the number of substitutions per site.

AM923), but in less than 20% of the isolates in clades B and C, were selected for further analysis (S7 Table).

The genes encoding these 63 CDSs were mostly scattered across the AM923 chromosome rather than organised into operons, with two notable exceptions. The first locus, at nucleotide positions 4462577–4466223 and upstream of the 6-phosphogluconolactonase gene (*plg*), was the large, small and cytochrome subunits of the fructose dehydrogenase (*fdhLSC*). The second locus, at nucleotide positions 919995–926406 and upstream of the nickel/cobalt transporter gene (*hoxN*), contained putative genes that encode proteins involved in the control of urea transport and degradation, *ureABCEFGD_utp*, (Fig 5). The presence of urease activity in strain AM923 and other strains positive in the *ure*B PCR assay was confirmed by phenotypic testing, which detected a positive reaction on urea agar slopes after 48 hours of incubation at 37°C (S2 Fig).

Among the 542 complete and partial *Serratia* spp. genomes analysed, the protein sequences for the urea transporter, the urease accessory proteins UreD, UreE, UreF and UreG, and the

**Table 2. Origin of isolates across different phylogenetic groups of *Serratia* spp. defined by the multiple alignment analysis of 14 housekeeping genes.**

| Origin | *Serratia marcescens sensu lato*[a] | | | Other *Serratia* species | Total |
|---|---|---|---|---|---|
| | clade A | clade B | clade C | | |
| Insect | 19 (39%) | 5 (6%) | 6 (1%) | 7 (9%) | 37 (6%) |
| Plant, Environment | 9 (18%) | 52 (58%) | 14 (3%) | 35 (47%) | 110 (18%) |
| Human, Blood, Hospital | 10 (20%) | 19 (21%) | 361 (90%) | 2 (3%) | 392 (64%) |
| Other or No Information | 11 (22%) | 14 (16%) | 22 (5%) | 30 (41%) | 77 (13%) |
| Total | 49 | 90 | 403 | 74 | 616 |

[a] Group of 542 genomes also containing strains identified as *S. ureilytica* (n = 3), *S. nematodiphila* (n = 6) or undefined *Serratia* spp. (n = 39).

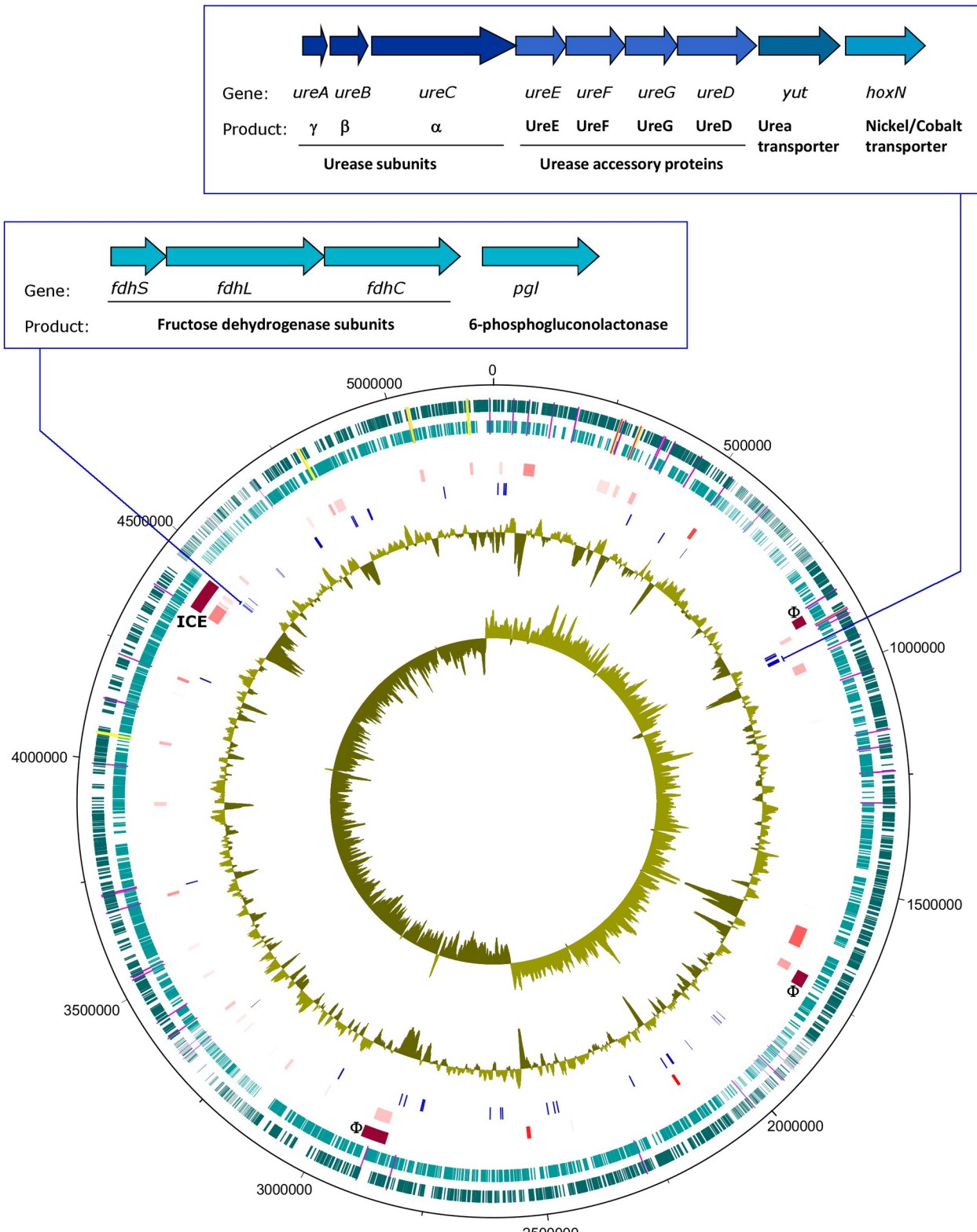

**Fig 5. Genome map of strain AM923.** This map displays loci that were putatively acquired horizontally and CDSs predominantly found in the genomes of *Serratia* spp. isolated from insects. Features positioned in concentric circles, from outer to inner tracks as follows: 1–2, dark teal, forward CDSs; light teal, reverse CDSs; mauve, tRNA; yellow, rRNA; 3, dark red, predicted prophages (Φ), Integrative Conjugative Transposon (ICE); 4, red shades, putative HGT region predicted by Alien Hunter; 5, blue, CDSs predominantly found in insect isolates; 6, G+C%; 7, G+C skew. Inset, map of urease locus (top) and fructose dehydrogenase locus (bottom).

urease subunits alpha, beta and gamma were found in all clade A isolates. In contrast, these genes were largely absent from the isolates in clades B and C, with only 16% and 9%, respectively, of genomes carrying the entire locus. This indicated that an assay specific for a urease operon sequence could be used for detection of entomopathogenic strains of *Serratia* spp. At the DNA level, systematic searches for the complete urease locus genes in all *Serratia* spp. genomes, using the AM923 sequences as Blastn queries, detected matching sequences in all members of clade A (including 19 insect isolates), 8/90 members of clade B, and 36/403 members of clade C. The matching clade C sequences included 4 insect isolates, strain AM1004 from Melbourne Zoo and 3 highly similar isolates from *Anopheles stephensi*, along with 32 human, plant or environmental isolates.

Since urease genes are commonly found in insect-derived isolates of *S. ureilytica* and *S. marcescens* and may contribute to the virulence of these strains, a specific PCR assay was designed based on the sequence of the *ure*B gene, which encodes the urease subunit beta. The expected 77 bp product was amplified from the genomic DNA of strain AM923 (S4 Fig). In addition, the sequence of the *nuc*A gene, which was conserved in 100% of the *Serratia* spp. genomes analysed, was used to design a positive control PCR assay that was shown to amplify a 67 bp fragment from the same extracted DNA samples (S4 Fig).

Despite some sequence divergence within the urease operon, *in silico* testing suggested that the *ure*B PCR primers were predicted to anneal with 83 of the 93 target sequences, with 0–2 mismatches in the first 6 bases and no mismatches in the last 13 positions of the forward primer, and no mismatches in the reverse primer. The *ure*B PCR assay was tested on a collection of *Serratia* spp. isolates from insect and non-insect hosts. The assay was able to detect most suspect entomopathogenic strains from Melbourne Zoo, as well as two isolates that had rep-PCR fingerprint profiles similar to that of AM923. Products were not amplified from other isolates from domestic animals (S2 Fig).

### Pathogenicity of *S. ureilytica* AM923 in an insect model

To assess the virulence of *S. ureilytica* for insects, European honey bees (*Apis mellifera)* were inoculated with a live culture of strain AM923 or *E. coli* K12, and mortality was monitored over time. Kaplan-Meier survival curve analysis revealed a marked difference between strain AM923 and the negative control *E. coli* K12, with cumulative mortalities of 40% and 5%, respectively, after 8 days (Fig 6). Log-rank test P values demonstrated a significant difference in mortality rates between the groups of bees exposed to AM923 and *E. coli* K12 ($P < 0.05$) or sterile medium ($P < 0.05$). In contrast, the mortality rates in the latter two groups did not differ significantly.

To address Koch's postulates, randomly selected bees that died or survived within each group were examined for the presence of *Serratia* spp. by culture and PCR. The MacConkey agar cultures of abdomens of bees exposed to strain AM923 yielded varied amounts of growth, dependent upon whether they died or survived the experiment. A heavy growth of *Serratia* spp. was obtained from insects that died during the experiment while a light growth or no significant growth was obtained from insects that survived. The *nuc*A-specific PCR assay indicated the presence of *Serratia* spp. in all bees tested from this group, and the *ure*B-specific PCR assay strongly suggested that strain AM923 was present. In contrast, no *Serratia* spp. were detected by culture or PCR in the bees screened before the experiment or in any of the *E. coli* K12 exposed or negative control groups (S4 Fig).

### Discussion

Genomic sequence and phenotypic analyses identified the predominant strain (AM923) isolated from *D. australis* as *Serratia ureilytica*. This species was first isolated from a river in India

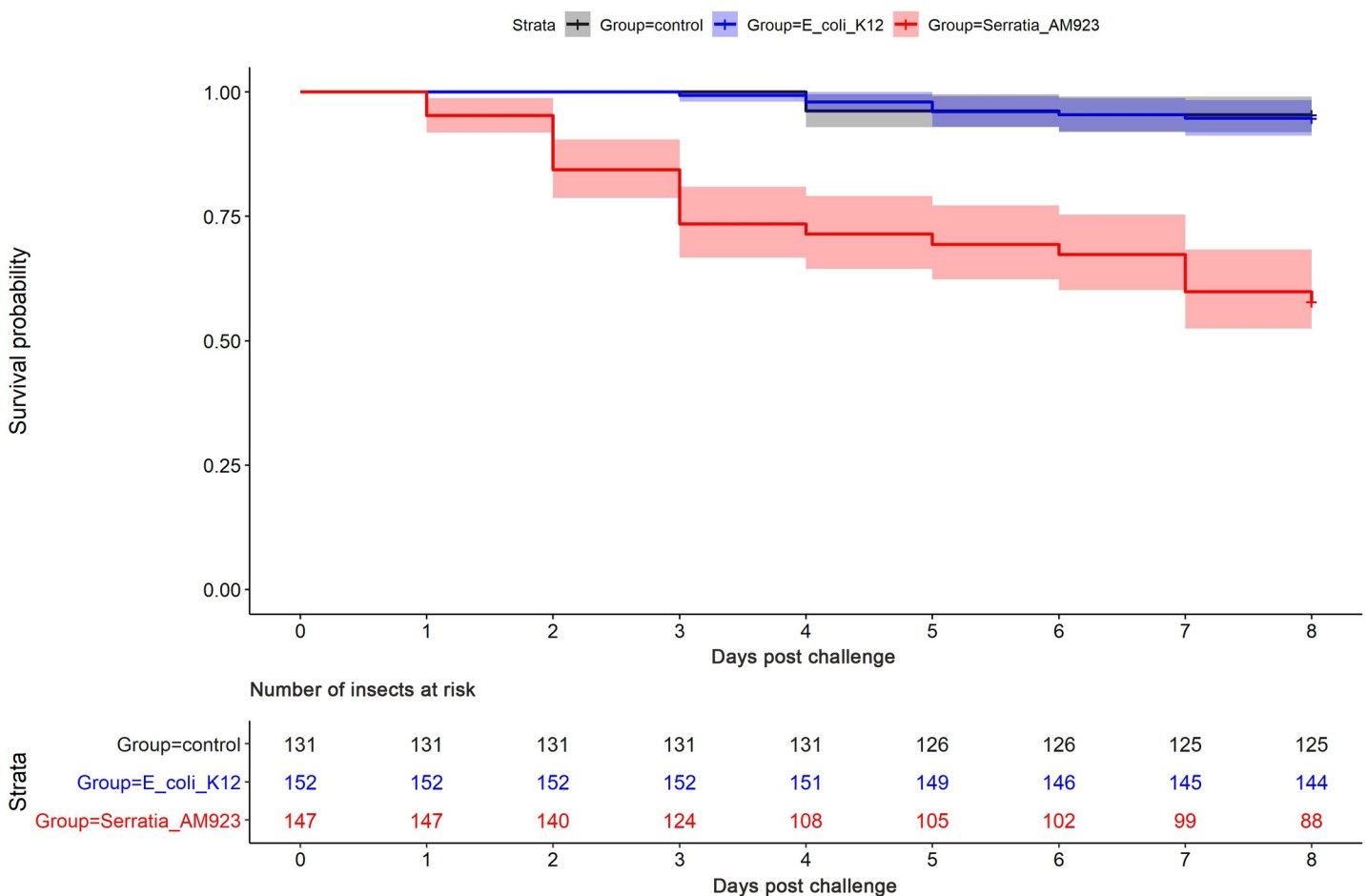

**Fig 6. Kaplan-Meier survival curves in honey bees challenged with a live culture of *S. ureilytica*.** Strain AM923 (red), *E. coli* K12 (blue), or exposed to sterile PBS and the sucrose/water mix as uninoculated negative controls (dark grey). Semi-transparent zone indicates a 95% confidence interval.

[43] and shown to be resistant to extreme environmental conditions [44]. It has also previously been identified in the gut microbiota of honey bees [45]. To our knowledge, this is the first report of this species as a potential insect pathogen.

Although *Serratia* spp. can be present in the environment, it is unlikely that the isolates recovered from the affected insects were culture contaminants, because *Serratia* spp. was isolated in pure culture on non-selective medium from the majority of haemolymph samples. Histopathological examination revealed disseminated inflammatory lesions frequently containing Gram negative rods, further suggesting that bacterial colonisation had occurred prior to death. The presence of only two *Serratia* pulsotypes, with the majority of isolates belonging to one type, suggests that the most likely reason for isolation of the *Serratia* sp. was not opportunistic infection, which would be expected to result in isolation of diverse strains of *Serratia* spp., but rather that a group of specific, closely related strains were able to colonise insects and that these pathogenic strains caused most of the mortalities.

Other bacterial species, including *Pseudomonas* spp. and *Proteus* spp., were also found in dead *D. australis*, but their relative contribution to the disease compared to *Serratia* spp. remains unclear. Furthermore, while the role of viruses and fungi in disease in captive *D. australis* cannot be discounted, no evidence has been found of non-bacterial infectious agents

associated with significant mortalities despite extensive histopathological and microbiological investigations over several years at Melbourne Zoo.

While the base-line mortality rate in the Melbourne Zoo population of *D. australis* is documented, the natural longevity of these insects is unknown. It is likely that host factors, such as poor genetic diversity resulting from the population bottleneck experienced by *D. australis*, as well as environmental factors such as hygiene and stocking densities in the captive environment, interact with pathogens and play important roles in the epidemiology of disease in *D. australis*. Nevertheless, isolation of *Serratia* spp. from the haemocoel suggests members of this species are important causal agents of disease and mortality in captive *D. australis*.

The phylogenetic analyses of this study raise questions about the phylogeny and taxonomy of the genus *Serratia*. While most of the *Serratia* spp. associated with insect infections included in this study were originally submitted to the NCBI sequence databases as *S. marcescens*, our analysis suggests that several of these isolates may in fact be *S. ureilytica*. For clarity, our results could be re-interpreted by including the *S. marcescens*, *S. nematodiphila* and *S. ureilytica* genomes into a "*S. marcescens sensu lato*" group and renaming (or re-ranking) the strains in individual clades. However, the taxonomic reclassification of publicly available genomes would require a thorough phenotypic and genotypic analysis beyond the scope of this study. Therefore, the original classification provided in the NCBI GenBank files were maintained.

A previous comparative genomic study of the genomes of *S. marcescens* Db11, a pathogenic isolate from *D. melanogaster*, and a multidrug resistant human isolate (SM39), found 3970 genes were conserved across both strains [46]. More recently, the relatedness of complete and unfinished genomes of *S. marcescens* from human nosocomial infections (22 strains), the environment (17 strains) and a limited number of insect sources (4 strains) were explored in a pangenome analysis [47]. While the human-associated isolates were clearly separated from the environmental isolates, the position of the four insect isolates within these groups was more ambiguous.

Our results confirm and expand upon these previous studies by including more insect associated *Serratia* spp. genomes in the analyses. Most strains associated with insects, including the representative predominant strain *Serratia ureilytica* AM923 found in *D. australis*, belong to a clade distinct from those containing mainly human or environmental isolates. *Serratia marcescens* strain AM1004, representative of a group less frequently isolated at the Melbourne Zoo and not persistent in the captive insect population, was found within the human-associated clade C, suggesting a separate origin, and possibly a more minor role of this subgroup in the mortalities. This exemplifies the complexity of the increased mortalities and demonstrates the importance of genome analysis to accurately describe the epidemiology of such events.

Entomopathogenic strains of *Serratia* spp. usually cause disease in a wide range of insects by entering the haemocoel from the gut, causing septicaemia and death [48]. Other aspects of the virulence of the organism during insect infections are largely unknown. Previous work to screen a mini-Tn5 transposon mutant library in *S. marcescens* Db11 found 23 mutants with attenuated virulence for *Caenorhabditis elegans*, and 9 of these 23 mutants had reduced virulence in *D. melanogaster*, including mutants defective in iron transport, haemolysin production and LPS biosynthesis [49]. Chitinases and the urease encoded by strains of *S. marcescens* isolated from *A. stephensi* were also predicted to be virulence factors [50].

Urease production is rarely observed in *S. marcescens* [7], but is a defining characteristic of *S. ureilytica*. Two insect-associated strains of *S. marcescens*, ano1 and ano2, isolated from the midgut of the mosquito *A. stephensi*, were found to possess a functional urease operon, and their comparison with 101 publicly available *Serratia* genomes revealed that only 12 other genomes encoded a urease [50]. In our study, all of the *Serratia* spp. genomes of clade A, which contains most of the insect-associated isolates, including AM923, and all 6 insect isolates from clade C, including AM1004, carry a complete urease operon, in contrast with most genomes from the

environmental and human isolates in clades B and C. The evolutionary origin and the function of the operon in *S. marcescens* strain AM1004 is unclear and will be explored in further work.

A functional urease operon (*ure*ABCEFGD) is also found in *Yersinia pseudotuberculosis*, an environmental member of the *Enterobacteriaceae* with entomopathogenic potential, but not in its evolutionary derivative, the flea-borne and flea-transmitted mammalian pathogen *Yersinia pestis*. Indeed, the mortality rates in two flea species, *Xenopsylla cheopis* and *Oropsylla montana*, fed a blood meal containing *Y. pseudotuberculosis* was higher than in those fed with *Y. pestis* [51]. Insects fed purified canatoxin, a variant form of bacterial urease contained within the seeds of the Jack Bean plant (*Canavalia ensiformis*), also experienced a high rate of mortality [52]. The effects of both *Y. pseudotuberculosis* and purified canatoxin can be abolished by the administration of the urease inhibitor, para-benzoquinone [51]. It is therefore possible that the urease found in AM923 and AM1004 may play an important role in their pathogenicity for *D. australis*. This question will be addressed in future studies.

*D. australis* is an endangered species, so it was not possible to optimise an experimental infection model in this species. Therefore, we sought to develop an alternative infection model to assess the virulence of strain AM923. Infection models using *C. elegans* and *D. melanogaster* have been used previously to assess the virulence of *Serratia* spp. [17, 48, 49]. However, more recently, the strains KZ-19, sicaria Ss1 and Db11, which we found to be genetically related to *S. ureilytica* AM923, were shown to kill bees whether they were fed, immersed in or injected with the different strains [18].

Our results suggest that a bee model of infection using the immersion protocol is a reliable, rapid and relatively easy method to test the virulence potential of strains isolated from *D. australis*. Koch's postulates were partially confirmed, as both culture and PCR suggested *S. ureilytica* was isolated only from dead insects in the *S. ureilytica* AM923 treatment group.

Finally, we propose several genomic targets upon which to base development of detection tools specific for entomopathogenic *Serratia* spp. found not only at Melbourne Zoo, but other zoos and captive insect breeding facilities. These molecular tools could be used to screen frass prior to the release of insects into their native habitat. Furthermore, phylogenetic investigations of *S. ureilytica* and *S. marcescens* will facilitate further understanding of the risk these bacteria pose to the reintroduction program for a critically endangered insect species.

## Supporting information

**S1 Fig. An example of the cup cage used to contain honey bees during the virulence assay.** Adapted from the Sistema KLIP IT™ Utility Collection round container with strainer, 0.7 L with transfer pipette bulbs inserted to supply water and 50% sucrose solution.
(TIF)

**S2 Fig. Rep-PCR fingerprinting analysis of *Serratia* spp. isolates.** A subset of Lord Howe Island stick insect isolates from Melbourne Zoo in 2017 and 2019, indicated by the red hashed box, displayed a rep-PCR fingerprint highly similar to *Serratia ureilytica* AM923. Dendrogram built with GelJ using Pearson curve-based similarity coefficients with the UPGMA linkage method. Isolation year, strain number, host and results *of* PCR assays (*ure*B and *nuc*A) and the urease production phenotypic test are indicated for each lane; n/d: no data. MZ: Lord Howe Island stick insect isolates from Melbourne Zoo. Molecular weight marker: HyperLadder™ 1kb (Meridian Biosciences).
(TIF)

**S3 Fig. Phylogenetic analysis of concatenated alignments of 8 core genes from 650 *Serratia* spp. genomes.** A mid-rooted tree was built from 3069 positions of 212 non-identical

sequences, including 169 *S. marcescens*, *S. nematodiphila* and *S. ureilytica* isolates, with RAxML using a GTR + Gamma model. Bootstrap analysis was performed on 100 replicates, and support values >50% are represented by circles with diameters proportional to the value. The Melbourne Zoo isolates AM923 and AM1004 are indicated on the outside ring. The scale bars indicate the number of substitutions per site.
(TIF)

**S4 Fig. Detection of *Serratia* spp. in honey bees challenged with a live culture of *S. ureilytica* AM923.** Two PCR assays were conducted to confirm detection of *Serratia* spp. in bees exposed to AM923, *E. coli* K12, or 1:1 mixture of sterile PBS and the sucrose/water mix (control). Arrows indicate the expected amplicon on the 3% agarose gel. The nature and time of the event is indicated for each bee. Dead: the individual was found dead during the experiment; Survivor: the individual was euthanised at the conclusion of the experiment; Pre-experiment: honey bee collected from the hive before inoculation; NTC: no template control. Molecular weight marker: HyperLadder™ 1kb (Meridian Biosciences).
(TIF)

**S1 Table. Details of the 131 *Serratia* spp. used in REALPHY phylogenomic analysis.**
(DOCX)

**S2 Table. Features of the 708 partial or complete *Serratia* genomes used for phylogenomic and pan-genome analyses.**
(DOCX)

**S3 Table. *Serratia* spp. isolated from the insect environment (water, nest boxes, frass and floor samples).**
(DOCX)

**S4 Table. Histopathological analysis of a subset of 25 dead insects.**
(DOCX)

**S5 Table. Summary of PFGE pulsotypes and rep-PCR profiles of *Serratia* spp. isolated from insect haemolymph and the insect environment.**
(DOCX)

**S6 Table. Fast ANI analysis to compare strains AM923 and AM1004 with reference strains of *Serratia* spp.**
(DOCX)

**S7 Table. CDSs present in all insect isolates from clade A and fewer than 20% of isolates from clades B and C.**
(DOCX)

**S1 Raw images.**
(PDF)

## Acknowledgments

We would like to acknowledge the work of staff at Melbourne Zoo and Zoos Victoria; the zoo-keepers, nurses and veterinarians for their assistance in the necropsies, sample collection, record-keeping, husbandry skills and overall dedication to the cause of the Lord Howe Island stick insect and the Zoos Victoria Wildlife Conservation and Science team for their support of this project. We would also like to acknowledge the work of Christine Bayley and the staff at Gribbles Veterinary Pathology for their expertise and assistance with the pathological investigations.

## Author Contributions

**Conceptualization:** Joanne L. Allen, Christina Cheng, Michael Lynch, Helen K. Crabb, Jean-Pierre Scheerlinck, Glenn F. Browning, Marc S. Marenda.

**Data curation:** Joanne L. Allen, Nicholas P. Doidge, Christina Cheng, Michael Lynch, Rhys Bushell, Marc S. Marenda.

**Formal analysis:** Joanne L. Allen, Nicholas P. Doidge, Christina Cheng, Helen K. Crabb, Glenn F. Browning, Marc S. Marenda.

**Funding acquisition:** Joanne L. Allen, Michael Lynch, Glenn F. Browning.

**Investigation:** Joanne L. Allen, Nicholas P. Doidge, Christina Cheng, Michael Lynch, Rhys Bushell, Glenn F. Browning, Marc S. Marenda.

**Methodology:** Joanne L. Allen, Nicholas P. Doidge, Christina Cheng, Jean-Pierre Scheerlinck, Rhys Bushell, Glenn F. Browning, Marc S. Marenda.

**Project administration:** Joanne L. Allen, Nicholas P. Doidge, Christina Cheng, Michael Lynch, Rhys Bushell.

**Resources:** Joanne L. Allen, Jean-Pierre Scheerlinck, Rhys Bushell, Glenn F. Browning, Marc S. Marenda.

**Software:** Nicholas P. Doidge, Helen K. Crabb, Rhys Bushell, Marc S. Marenda.

**Supervision:** Joanne L. Allen, Michael Lynch, Glenn F. Browning, Marc S. Marenda.

**Validation:** Joanne L. Allen, Nicholas P. Doidge, Christina Cheng, Marc S. Marenda.

**Visualization:** Joanne L. Allen, Nicholas P. Doidge, Marc S. Marenda.

**Writing – original draft:** Joanne L. Allen, Nicholas P. Doidge, Marc S. Marenda.

**Writing – review & editing:** Joanne L. Allen, Nicholas P. Doidge, Christina Cheng, Michael Lynch, Helen K. Crabb, Jean-Pierre Scheerlinck, Glenn F. Browning, Marc S. Marenda.

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
