## [Decision Letter · Decision Letter 0]

22 Dec 2021

PONE-D-21-36966An entomopathogenic strain of *Serratia ureilytica* is associated with mortalities in a captive colony of the critically endangered phasmid *Dryococelus australis*PLOS ONE

Dear Dr. Allen,

Thank you for submitting your manuscript to PLOS ONE. After careful consideration, we feel that it has merit but does not fully meet PLOS ONE’s publication criteria as it currently stands. Therefore, we invite you to submit a revised version of the manuscript that addresses the points raised during the review process.

We look forward to receiving your revised manuscript.

Kind regards,

Chih-Horng Kuo, Ph.D.

Academic Editor

PLOS ONE

Journal Requirements:

Additional Editor Comments:

Please check the PLOS ONE Criteria for Publication (https://journals.plos.org/plosone/s/criteria-for-publication) when planning the revision.

Specifically, the experiments need to be described in sufficient details and the conclusions (including the title) must be supported by the data presented.

Reviewers' comments:

Reviewer's Responses to Questions

**Comments to the Author**

1. Is the manuscript technically sound, and do the data support the conclusions?

Reviewer #1: Yes

Reviewer #2: Yes

Reviewer #3: Partly

2. Has the statistical analysis been performed appropriately and rigorously? 

Reviewer #1: Yes

Reviewer #2: No

Reviewer #3: I Don't Know

3. Have the authors made all data underlying the findings in their manuscript fully available?

Reviewer #1: Yes

Reviewer #2: Yes

Reviewer #3: Yes

4. Is the manuscript presented in an intelligible fashion and written in standard English?

Reviewer #1: Yes

Reviewer #2: Yes

Reviewer #3: Yes

5. Review Comments to the Author

Reviewer #1: My only suggested edit is to define MLST on line 72. Otherwise, the manuscript is excellent, the English level is fluent, and I have no further edits to recommend.

Reviewer #2: The title of the study is “ An entomopathogenic strain of Serratia ureilytica is associated with mortalities in a captive colony of the critically endangered phasmid Dryococelus australis”. However, it did not reflect the genetic characterization of the Serratia strains isolated from Dryococelus australis and its phylogenetic relationship with Serratia genus. I suggest that the title be changed to a title that shows all the work done in the present work..

The objectives of this research were: genome sequencing of two representative strains and they were compared with genomes of Serratia spp. Publicly available, including entomopathogenic strains. The virulence of a selected isolate was evaluated in experimentally infected honey bees (Apis mellifera).

Hemolymph samples were collected from necropsies of D. australis within the Melbourne Zoo colony that died or were found moribund and euthanized. Serratia, Pseudomonas or Aeromonas and Proteus were isolated from the samples. In addition, environmental samples [nest boxes, floors, drinking water and frass (insect excrement)] were collected.

Molecular typing was performed for Serratia strains based on PFGE and Fingerprinting rep-PCR. For Illumina sequencing, it was performed from two representative strains:: S. ureilytica AM923 and S. marcescens AM1004. The nucleotide sequencing of each strain was analyzed and prophages, genomic islands, antimicrobial resistance genes, and virulence were identified. On the other hand, a phylogenetic and pangenomic analysis was carried out against representative genomes of the genus Serratia downloaded from databases.

The virulence assay was carried out on bees; and finally, the nuclease (nucA) and urease beta subunit (ureB) genes were amplified.

I don't understand why the sequence of strain AM923 was chosen as the main reference genome and an in-depth analysis was performed, while strain AM1004 was used for comparative purposes. In this case, although the genotype of the strain AM923 was the predominant one in relation to AM1004, both genotypes were found in the samples. I think that both strains should be studied with the same interest, for example, both are reported in other Zoo. This means that both genotypes are circulating in the region and it is not known whether they can infect other types of hosts that are equally susceptible to infection by both types of strains.

It would also be desirable to perform virulence tests for strain AM1004.

Regarding the sequence of the urease operon that could be used for the detection of entomopathogenic strains of Serratia, a statistical analysis is recommended to validate this statement.

The presence of one or two pulsotypes of Serratia strains associated with the infection or death of Dryococelus australis, suggests an epidemic outbreak in the D. australis colony at Melbourne Zoo by one or two strains. It would be important to have the exact date on which the infection occurred in the insects and to relate them to the genetic characteristics of the strains. In addition, to determine the source of infection, it would be good to point out the origin of the environmental samples [nests, floor surfaces, drinking water and frass (insect excrement)] that were contaminated with Serratia, and sequence it, to compare the genome of the Serratia strain isolated from the environmental sample with the genome of the Serratia strain isolated from samples of insect hemolymph.

In general, the work is good, however I feel that the main objective is missing, which is the characterization of Serratia spp. strains associated with infection and death of the D. australis colony at Melbourne Zoo.

The authors focused on determining the phylogeny and taxonomy of Serratia genus.

In conclusion, I recommend that the authors rethink the title of the article and the objectives.

Reviewer #3: Paper outlines the isolation of insect pathogens and follows the route of species identification using in silico approaches.

To provide context it would be of value to expand on the insect raring conditions, the food source etc and how this in turn may relate to their natural habitat. From here can then associate with Serratia species

Though I see the rational for the use of Bees as a proxy for stick insects, which in turn may explain the use of 37 Celsius microbial growth parameters. Assessing an insect from the same family might be more preferable, more so as many insect pathogens are host specific

In relation to the disease are there any photos of the diseased insect. I note histology mentioned L103, some images would be of value - where is this data .

Table S4 lists observed plaques wherein some photos would help or for that matter more detailed bioassay data, host range and dose response data would be of value

In general greater detail could be provided in Figure legends

I feel that as written the manuscript is too focused on use of in silico approaches to define species and needs to be bolstered in mode of action and bioassay data more so to provide relevance to the study and for that matter the phasmid. In silico analysis can then assess for presence of insect active enzymes/toxins etc . In addition to this more information on how this study could help with raring phasmids in the discussion would be of value

Virulence assay

I am left unsure on the definition of carriage or a timeframe (line 465 denotes 6 days), I am left asking several questions :-

were they surface sterilised prior to isolation

was a dilution series undertaken prior platting

what is the dominance of the isolated bacteria relative to the other isolated bacteria -if any. In this context Table 1 lists some numbers. In this respect n=127 does this mean colonies total from an insect ? or something else - please clarify and provide a quantifiable measure , a foot note on definition of pure and mixed would be useful

L211 I am unsure of this assay it read as a direct immersion of a bee in a solution of bacteria based on the provide information it is hard to know how the bacteria gets ingested via this technique , perhaps using a sucrose bacteria solution applied to a dental wick might be more apt

Kochs postulates i.e. would be nice to see i.e. re feed the suspected bacteria back to the insect species it was isolated from to see affect again . I appreciate a rare insect so perhaps a closely related species might act as a more suitable proxy than the honey bee

In addition it appears the bacteria were reisolated and validated through PCR were the insects surface sterilised prior to this step?? I am mainly concerned with PCR sensitivity and its ability to differentiate live verse dead (DNA) cells . In this context platting out is a likely more applicable method more so if we assume that a pathogen might multiply and dominate the microbial population. It would be good to have a quantifiable measure eg from each cadaver ## bacteria were isolated

Phylogenies are not my expertise , this study are restricted based on species where sequenced genomes are available, this is okay but I think a MLST /16S encompassing all Serratia species (Type strains – of which data is available ) would help complement the extensive genome based phylogenies and help the reader better understand where the Serratia strain fits and of course interpretation of the data.

The authors also allude to the role of chitinases etc L559 in other systems but I am unsure if they assessed this in the genomes sequenced in this study. L553-560 are pertinent points and leaves the reader wondering what accessory virulence determinants are in the assessed Serratia genome - in this context a preliminary genome assessment of Haemolysin chitinase etc would be of value . This is possibly more relevant than Urease

Fig S2 is nice given a finite set of Serratia species (due to genome access limitations), please reference genome accession numbers. This is also applicable for other figures and Tables (I might have missed this information)

Of interest in the discussion the authors acknowledge and discuss many of the issues I have raised such as use of bee model etc

Minor comments are as per below

L37 Based on in silico analysis ? a urease operon …..

L52 could list food source

L57 is bacilli the correct term please check is this relating to Serratia? might be rods? I am unsure

L61 -62 why have we singled out these the Serratia specie?

L102 rational for using SBA and MAC? latter L118 LB then L130 TSB then used? I note MAC is used to selectively isolate Gram-negative and enteric bacteria

L103 is there a rational for 37C just curious might be of interest to understand the ambient temperature the phasmids insects are in

L118 perhaps mention cycloheximide is an anti fungal

L120 could get the 16s of the bacteria sequenced - in addition is there a predominant isolate

Figure 6 Time? I assume days post challenge please provide value, I am unsure what number at risk relates to.

L212 full genus for E coli

L213 has PBS been written in full and components provided?

Table S7 I am unsure of the relevance of this Table

L223 what is a cup cage perhaps dimensions could be provided?

L319 could list the program used prior to … Using #### the chromosome was predicted

L489 what is defined as a harsh environment

Fig S1 mention why a subset of isolates are red hashed boxed. i.e. red hashed boxed denotes Lord H..

6. PLOS authors have the option to publish the peer review history of their article (what does this mean?). If published, this will include your full peer review and any attached files.

Reviewer #1: **Yes: **Matan Shelomi

Reviewer #2: No

Reviewer #3: No

---

## [Author Response · Author response to Decision Letter 0]

24 Jan 2022

We would like to thank the reviewers for their helpful comments and questions. Our detailed responses to the reviewers' remarks and questions are covered in the document 'Response to reviewers'.

---

## [Decision Letter · Decision Letter 1]

22 Feb 2022

PONE-D-21-36966R1Genomic characterisation of an entomopathogenic strain of Serratia ureilytica in the critically endangered phasmid Dryococelus australisPLOS ONE

Dear Dr. Allen,

Thank you for submitting your manuscript to PLOS ONE. After careful consideration, we feel that it has merit but does not fully meet PLOS ONE’s publication criteria as it currently stands. Therefore, we invite you to submit a revised version of the manuscript that addresses the points raised during the review process.

We look forward to receiving your revised manuscript.

Kind regards,

Chih-Horng Kuo, Ph.D.

Academic Editor

PLOS ONE

Additional Editor Comments:

Reviewer #3 provided additional comments to be addressed. I agree with the concern that the Discussion section is difficult to read, perhaps dividing into sub-sections with clear headings would help.

Reviewers' comments:

Reviewer's Responses to Questions

**Comments to the Author**

1. If the authors have adequately addressed your comments raised in a previous round of review and you feel that this manuscript is now acceptable for publication, you may indicate that here to bypass the “Comments to the Author” section, enter your conflict of interest statement in the “Confidential to Editor” section, and submit your "Accept" recommendation.

Reviewer #2: All comments have been addressed

Reviewer #3: (No Response)

2. Is the manuscript technically sound, and do the data support the conclusions?

Reviewer #2: Yes

Reviewer #3: Partly

3. Has the statistical analysis been performed appropriately and rigorously? 

Reviewer #2: Yes

Reviewer #3: I Don't Know

4. Have the authors made all data underlying the findings in their manuscript fully available?

Reviewer #2: Yes

Reviewer #3: Yes

5. Is the manuscript presented in an intelligible fashion and written in standard English?

Reviewer #2: Yes

Reviewer #3: Yes

6. Review Comments to the Author

Reviewer #2: (No Response)

Reviewer #3: The paper is much improved, specifically the introduction and the results, however the discussion lacks a clear focus and highlighting the outcomes of the study --and could be reworked . The responces in places are good but I might suggest that they accordingly update the manuscript with the information in the response provided

A photo of a diseased insect referenced in Bayley however I am left unsure if isolates from the Bayley paper were included in this study? if so please highlight this, if not this would further warrant the inclusion of a photo/histology from where it could be compared to that of Bayley et al

L120 I am still left unsure of what insects are feeding on - what is the plant material is it fresh a live plant etc??

L125 right LB in full first time mentioned

L280 perhaps put a descriptor on what the other groups are

L343 344 list % ANI value after each species name in brackets

L464 for future reference you could incorporate through primer synthesis in generic primer base at points of difference e.g. a N or a Y etc

L497 what is the extreme variable salinity , temp or other?? please update as per responce

Discussion -reads as disjointed and seems to focus on negatives detracting from the reading. I believe a discussion should be written in a positive context and highlighting key results

L503 do we mean isolated as a pure culture ? ….L504-505 could be deleted as detracts from the prior sentence

L505-506 are there pictures of the histology and I would possibly shift sentence to latter on after you have discussed the bacteria

L514 paragraph. I still think we need to be quantitative did Pseud and Prot dominate the % microbes isolated from these insects

L532 clarify who submitted the sequences to NCBI , if they -are from this study the sentence can be removed and leave it to NCBI to reassign the species , please clarify

L533-L539 may irk some readers could reduce text here and often good to cross reference/compare to a type strains, we can assume there is a lot of rubbish in the NCBI database,

L534 what do we mean by reinterpreted I might have suggested that these strains be included further strengthening the article

L541 this paragraph seems to have no value , what is its context and how does it relate to the results in this study

L606 what are these genomic targets

7. PLOS authors have the option to publish the peer review history of their article (what does this mean?). If published, this will include your full peer review and any attached files.

Reviewer #2: **Yes: **Rosario Morales-Espinosa

Reviewer #3: No

---

## [Author Response · Author response to Decision Letter 1]

4 Mar 2022

Thank you for providing us with the opportunity to submit a revised version of our manuscript. 

Our response to Reviewer #3's comments are addressed in the attached Response to Reviewers document.

---

## [Editor Report · Decision Letter 2]

11 Mar 2022

Genomic characterisation of an entomopathogenic strain of *Serratia ureilytica* in the critically endangered phasmid *Dryococelus australis*

PONE-D-21-36966R2

Dear Dr. Allen,

We’re pleased to inform you that your manuscript has been judged scientifically suitable for publication and will be formally accepted for publication once it meets all outstanding technical requirements.

Kind regards,

Chih-Horng Kuo, Ph.D.

Academic Editor

PLOS ONE
---

## [Editor Report · Acceptance letter]

29 Mar 2022

PONE-D-21-36966R2 

Genomic characterisation of an entomopathogenic strain of *Serratia ureilytica* in the critically endangered phasmid *Dryococelus australis*

Dear Dr. Allen:

I'm pleased to inform you that your manuscript has been deemed suitable for publication in PLOS ONE. Congratulations! Your manuscript is now with our production department. 

Kind regards, 

on behalf of

Dr. Chih-Horng Kuo 

Academic Editor

PLOS ONE